# Exciton Manifolds in Highly Ambipolar Doped WS_2_

**DOI:** 10.3390/nano12183255

**Published:** 2022-09-19

**Authors:** David Otto Tiede, Nihit Saigal, Hossein Ostovar, Vera Döring, Hendrik Lambers, Ursula Wurstbauer

**Affiliations:** 1Institute of Physics, University of Münster, Wilhelm-Klemm-Str. 10, 48149 Münster, Germany; 2Center for Soft Nanoscience (SoN), University of Münster, Busso-Peus-Straße 10, 48149 Münster, Germany

**Keywords:** electrolyte gating, transition metal dichalcogenides, excitons, exciton binding energy, spectroscopic imaging ellipsometry, dielectric functions

## Abstract

The disentanglement of single and many particle properties in 2D semiconductors and their dependencies on high carrier concentration is challenging to experimentally study by pure optical means. We establish an electrolyte gated WS${}_{2}$ monolayer field-effect structure capable of shifting the Fermi level from the valence into the conduction band that is suitable to optically trace exciton binding as well as the single-particle band gap energies in the weakly doped regime. Combined spectroscopic imaging ellipsometry and photoluminescence spectroscopies spanning large n- and p-type doping with charge carrier densities up to 10${}^{14}$ cm${}^{-2}$ enable to study screening phenomena and doping dependent evolution of the rich exciton manifold whose origin is controversially discussed in literature. We show that the two most prominent emission bands in photoluminescence experiments are due to the recombination of spin-forbidden and momentum-forbidden charge neutral excitons activated by phonons. The observed interband transitions are redshifted and drastically weakened under electron or hole doping. This field-effect platform is not only suitable for studying exciton manifold but is also suitable for combined optical and transport measurements on degenerately doped atomically thin quantum materials at cryogenic temperatures.

## 1. Introduction

Two-dimensional materials beyond graphene have attracted enormous interest because of their potential for application in several areas including (flexible) nano-opto-/electronics, catalysis, solar energy conversion, (bio-)sensing, spin- and valleytronics [1,2,3,4,5]. A rich variety of atomically thin materials including metals, semimetals, semiconductors, insulators, superconductors and topological insulators exist. Semiconducting transition metal dichalcogenides (TMDCs) are van der Waals (vdW)-layered crystals of the form MX${}_{2}$ such as MoS${}_{2}$, WS${}_{2}$, MoSe${}_{2}$, WSe${}_{2}$. Those semiconducting TMDCs are the most explored two-dimensional materials post-graphene thanks to their availability, stability, ease of processing and interesting physical properties [6]. In particular, semiconducting TMDCs feature unique optical and optoelectronic properties such as an exciton-dominated optical response even at room temperature [7,8,9,10], doping-induced and presumably exotic superconductivity [11,12,13,14], fascinating spin and valley properties up to room temperature [2] and pronounced interaction physics [15,16,17].

Interactions are not only relevant to study electron correlation phenomena and emergent quantum phases, but can also have strong impact on the electronic band structure as well as on exciton physics and, hence, the optical properties. Reduced dimensionality together with reduced screening result in large exciton binding energies in the order of 0.5 eV in these semiconducting TMDCs [8,9,10]. There is an attractive Coulomb interaction between the bound electron and hole pair building an exciton and, as a consequence, the exciton binding energy can be manipulated with external stimuli such as strain [18], its dielectric environment [19,20], intense photoexcitation [21] or electron doping in field-effect structures [22,23]. Such external stimuli can cause band renormalization effects, counteracting the change in binding energy such that measured energies in optical interband experiments sensitive to the exciton ground states are only slightly changed with doping [24,25]. Due to the mutual impact of external stimuli on single-particle and excitonic properties, it is extremely difficult to disentangle them in all-optical experiments.

In a similar context, there is an ongoing debate in literature on the microscopic origin of the exciton manifold observed in photoluminescence (PL) spectra of monolayer (ML) WS${}_{2}$ for which the lowest interband transition spin-forbidden. The corresponding spectral features are assigned in different studies to recombination of direct excitons, momentum-forbidden and phonon-activated excitons, trion species (charged exciton), defect-related excitons and biexcitons [23,26]. A better understanding of the effect of doping on exciton formation, on electric field induced renormalization of electronic bands, electron–electron and electron–phonon interaction is of great interest for the interpretation and the control of optical properties of WS${}_{2}$ monolayers. In addition, high electron densities in the order of 10${}^{14}$ cm${}^{-2}$ are reported to result in the emergence of quantum phase transitions such as gate-induced superconductivity in semiconducting TMDCs [13,14] or even a light controllable electronic phase transition [27].

In order to study the electric field and doping induced impact on the band-gap, optical and emergent properties by all-optical means, an experimental platform is required that allows for an in-situ tuning of electric field and charge carrier density over a wide range from high electron to high hole doping compatible with optical absorption and emission experiments. Substitutional doping [28], chemical treatment [29] or surface evaporation of dopants [30] lacks in-situ tuning of the charge carrier density. The most suitable way is to embed WS${}_{2}$ ML in field-effect transistor (FET) structures using e.g., SiO${}_{2}$ or hBN layers as gate dielectric. However, the electric field strengths achievable with those structures are limited and typically not suitable to shift $E_{F}$ across the band gap for capacitance spectroscopy of the electronic bands and to inject high *n*- and *p*-type doping levels due to limitations of the gate dielectric. A successful strategy is the use of solid or liquid electrolyte gates allowing the injection of high charge carrier densities [31,32].

In this paper, we report on the successful implementation of solid electrolyte gated WS${}_{2}$ MLs devices suitable to tune the Fermi energy $E_{F}$ from the valence (VB) to the conduction band (CB) edge and to study the evolution of exciton manifolds in WS${}_{2}$ in presence of charge carriers spanning a large electron and hole doping range up to 10${}^{14}$ cm${}^{-2}$. This field-effect platform allows direct optical access to the VB and CB edges of WS${}_{2}$ MLs. At the same time, spectroscopic imaging ellipsometry (SIE) provides access to the exciton ground and excited (Rydberg) states, enabling the estimation of exciton binding $E_{bind}$ and the single-particle gap energy $E_{gap}$ and to trace their evolution while shifting the $E_{F}$ from VB to CB. Both show clear maxima for *E*${}_{F}$ located in the middle of band gap. The corresponding values constitute $E_{bind}\approx$ 500 meV and $E_{gap}\approx$ 2.5 ± 0.1 eV in agreement with reports in the literature [8,19]. Moreover, combined SIE and PL spectroscopies allow the study of the rich exciton physics including higher lying exciton transitions in WS${}_{2}$ and their dependencies on the charge carrier polarity and density up to 10${}^{14}$ cm${}^{-2}$. We show that the two most prominent emission bands are due to the recombination of momentum- and spin-forbidden charge neutral excitons that are activated by phonons. A third red-shifted emission line appears in the presence of excess charge carriers and is assigned to charged excitons [21,24]. For high *n*- and *p*- doping, the evolution of exciton manifolds as well as exciton dissociation (Mott transition) [33] is monitored. We discuss the rich PL spectra in the framework of different partially phonon activated charge neutral excitons, charged excitons, excitons dressed by a cloud of electrons or holes up to fully dissociated electron–hole pairs [17,26,34,35,36,37].

## 2. Materials and Methods

### 2.1. Sample Preparation and Introduction of Electrolyte Field Effect Structure

WS${}_{2}$ field-effect structures are prepared by micromechanical exfoliation and viscoelastic dry transfer of WS${}_{2}$ flakes (bulk crystals supplied by hq graphene) on top of pre-patterned glass substrates (single sided polished BOROFLOAT^®^ 33 borosilicate glass). The metal electrodes contacting the flakes and the electrolyte are patterned by evaporation of 5 nm chromium as adhesion layer followed by 80 nm gold through a shadow mask to avoid surface contamination from resist residues. As an electrolyte top gate, we use a solid polymer electrolyte consisting of poly-(ethylene oxide) and LiClO${}_{4}$ (ratio 8:1) dissolved in methanol and deposited by spin-coating using 8000 rpm followed by an annealing step on a hot plate. An optical micrograph of such a WS${}_{2}$ field effect device together with PL maps are shown in Appendix A. The fabrication process is optimized to obtain an homogeneous thin polymer electrolyte film with large domains important for optical measurements. For improved results, the thickness of a film for spectroscopic imaging ellipsometry (SIE) measurements shall be less than the wavelength of light in the media and the PE thickness was therefore kept in the order of 350 nm. The thickness of the electrolyte top gate was roughly doubled for PL measurements in order to enhance the long-term stability of the samples under illumination with high light intensities at room temperatures. Those thicker films are prepared by keeping the ion concentration constant. Only the amount of solvent with respect to the solid parts together with the spinning parameters are adjusted. In this way, the gate capacitance of the electric double layer transistor (EDLT) formed by the mobile ions on top of the WS${}_{2}$ layer is assumed to be independent from the film thickness. Nonetheless, minor variations are expected from sample to sample (see gate-dependent PL measurements on two different samples in Appendix A). The WS${}_{2}$-EDLT electrodes behave like a plate capacitor for a finite density of states in the WS${}_{2}$ sheet (meaning $E_{F}$ inside CB or VB). We refer to this situation as regimes II for charge carrier densities $<10^{13}$ cm${}^{-2}$ and regime III for densities $>10^{13}$ cm${}^{-2}$. The thickness of the double layer capacitor is in the order of very few nm resulting in an enormous gate capacitance per unit area. In regimes II and III, the capacitance of the investigated devices is approximated to $C_{g}\approx$ 1.5 μF/cm${}^{-2}$ with a distance within the EDL in the order of 2–3 nm [31]. This corresponds to a change in charge carrier density of roughly $1.25\times 10^{13}$ cm${}^{-2}$ per applied volt. This gate-induced charge carrier density has been quantified by μ-Raman measurements on MoS${}_{2}$-EDTL in a previous study [38]. We would like to note that in order to avoid surface contamination negatively impacting, particularly, the modelling of the SIE spectra, the gold electrode is not covered by an insulating film, but in direct contact with the electrolyte such that charge accumulation or depletion as well as surface oxidation and reduction of the gold can occur [39]. In turn, the circuitry consists of two capacitive elements with the capacitance of the semiconducting WS${}_{2}$-EDLT electrode beeing changed for $E_{F}$ inside $E_{gap}$ (regime I) and inside CB or VB (regimes II,III). It is not possible to use a reference electrode in our experiments since scattered and reflected light from such an electrode, in particular from the edges, would render meaningful SIE investigations nearly impossible. For these reasons, the utilized geometry does not allow for quantitative capacitance spectroscopy providing direct access to the band-gap $E_{gap}$ [40], but still allows for a qualitative capacitance spectroscopy indicating the gate voltage required to shift *E*${}_{F}$ to the CB and VB edges. With $E_{F}$ inside the CB or VB (regimes II and III), the WS${}_{2}$-EDLT can be well approximated as plate capacitor with the defined capacitance $C_{g}$. The impact from the Au-electrode interfacing the EDL on the estimated change in charge carrier density becomes negligible within the experimental uncertainty. Overall, more than 8 samples have been prepared and studied, and some of the samples have been intensely measured with several gate cycles. The displayed spectra and results are typical spectra achieved from those structures.

### 2.2. Electric Control and Functionality of the Electrolyte Gate

Electronic control over the gate potential during the optical measurements is realized by a source-measurement unit (Keysight Technologies). In order to minimize hysteresis effects, the top gate voltage is changed with a very slow scan rate of ≤3.3 mV/s${}^{-1}$ for PL and 2 mV/s${}^{-1}$ for SIE measurements, respectively. Leakage currents are monitored during the optical measurements to ensure stability of the gate. The gate voltage is swept at most between −13 V and 13 V. Even with this low scanning approach, well-known hysteresis effects due to low-mobile ions and (interfacially) trapped charges cannot be completely suppressed [40]. For this reason, the sweep direction and rate were kept constant in all measurements.

### 2.3. Spectroscopic Imaging Ellipsometry

Spectroscopic imaging ellipsometry (SIE) measurements are done with an imaging ellipsometer (accurion GmbH) using a supercontinuum white-light source with narrow tunable filters (both NKT Photonics GmbH) for illumination. The experiments are carried out at room temperature. In order to improve the stability of the field-effect structures during optical measurements, the samples are placed inside a home-built vacuum cell (vacuum p< 10${}^{-3}$ mbar) with windows suitable for SIE experiments in reflectance geometry using 55${}^{\circ }$ as angle of incidence. The incident light is guided through a polarizer for linear polarization and then through a compensator to prepare elliptically polarized light. The reflected light is directed through an ultra-low *NA* objective ($NA\approx 0.01^{\circ }$) to maintain nearly parallel light and imaged by an array detector after passing an analyzer. In a suitable coordinate system, the complex reflectance matrix is described by $\rho =r_{p}/r_{s}=\tan\Psi \cdot \exp i\Delta$, where ρ is the complex reflectance ratio, $r_{p}$ and $r_{s}$ are the amplitudes of the parallel (*p*) and orthogonal (*s*) components of the reflected light normalized to the amplitude of the incoming light, Ψ and Δ are the ellipsometric angles. To describe the SIE spectra of the field-effect structures and to extract the dielectric functions of WS${}_{2}$ ML in dependence of the gate voltage, a comprehensive multilayer model consisting of substrate, interlayer, WS${}_{2}$ and electrolyte layer each described by a suitable combination of Cauchy and/or Lorentzian and/or Tauc-Lorentz terms is developed and fit to the experimental data via regression analysis [41,42]. This regression analysis enables us to extract the dielectric functions of the WS${}_{2}$ layers for different gate voltages. To improve the accuracy, several regions on and off the WS${}_{2}$ layers have been investigated. By this approach, SIE measurements provide access to the dielectric function even if the imaginary part is largely suppressed by nearly fully screened excitons as it is the case for high doping densities. In this scenario, measurements of the absorption spectra from differential reflectance and extracting the dielectric functions by a Kramers–Kronig analysis is no longer possible [43].

### 2.4. Photoluminescence Spectroscopy

The μ-PL measurements are performed with a home-built microscope set-up using free beam optics. The samples are placed on x-y-z piezo actuators for precise positioning and are excited with the green light of a solid-state diode laser emitting at 532 nm. The excitation power is kept constant at around 50 μW. The light is focused on the sample using a 50× magnification objective (Mitutoyo) resulting in a spot size with a diameter of less than 2 μm corresponding to an excitation power density of about 16 kW/cm${}^{2}$. The emitted light is focused to the entrance slit of a single stage grating spectrometer (Princeton instruments) and the dispersed light is collected using a CCD. All optical measurements are carried out in vacuum (p< 10${}^{-3}$ mbar) at room temperature.

## 3. Results

### 3.1. Optically Detected Band Edges

The optical measurements on highly ambipolar doped WS${}_{2}$ monolayers are carried out on the optimized field-effect structures sketched in Figure 1a that are suitable for gate-voltage-dependent PL and SIE studies. The electronic double layer (EDL) provides high capacitance such that electric field tunability and charge carrier dependence of the absorption and emission spectrum over a wide range can be studied. The large geometrical gate capacitance allows us to shift the chemical potential μ and, hence, $E_{F}$ through the band gap from the CB to the VB of WS${}_{2}$ and vice versa as schematically depicted in Figure 1b. In this way, exciton species can be studied in absence and presence of free charge carriers with varying polarity and density over a wide range (see Figure 1b).

The relation between exciton binding energy $E_{bind}$, optical interband transition $E_{opt}$ and single-particle band gap is $E_{gap}=E_{opt}+E_{bind}$ as sketched in Figure 1b. This relation does not directly allow the determination of $E_{gap}$ and $E_{bind}$ from optical interband emission or absorption measurements. The application of an electric field via the gate voltage $V_{g}$ on the WS${}_{2}$ field-effect structure causes a shift of the chemical potential Δμ of the gate with respect to the electronic band structure and a shift of the electrostatic potential $\Delta \Phi =e\Delta n\left(p\right)/C_{g}$ with Δn(p) being the electrostatically induced change in electron (hole) densities and $C_{g}$ the gate capacitance [31,40]. The Fermi level $E_{F}$ of the semiconductor aligns with the electrochemical potential μ. Overall, we distinguish between three regimes regarding the position of $E_{F}$ (equivalently μ) relative to the band edges as indicated in Figure 1b. In regime I, the Fermi level $E_{F}$ of the semiconducting WS${}_{2}$ ML is located inside the single-particle band gap of WS${}_{2}$. Regime II and regime III correspond to $E_{F}$ inside the CB or VB causing high and very high charge doping, respectively.

For regime I, the effect of the gate voltage on the change of the charge carrier density Δn(p) is negligible since the density of states N(E) inside the band gap of a pristine, defect free semiconductor is vanishingly small and as a direct consequence $C_{g}$ becomes vanishingly small as well. Therefore, a change of the gate voltage in regime I causes a shift of μ with $e\Delta V_{g}\propto \Delta \mu$ allowing for a qualitative spectroscopy of the CB and VB edges of the semiconductor from the distinct change in the integrated PL intensities due to Fermi-edge singularities when $E_{F}$ is aligned to the CB or VB edges [44].

### 3.2. Emission Spectra in High-Doping Regime

In regime II, the chemical potential approaches the VB and CB band edges such that there are, in addition to thermally excited charge carriers, initial electrons (holes) electrostatically injected from the contact to the WS${}_{2}$ CB (VB). With further increase of the positive or negative gate voltage, the change of the electrostatic potential is no longer negligible since there is a high density of states $N_{E}$ inside the CB and VB of the semiconductor such that the shift of the electrostatic potential $\Delta \Phi =e\Delta n\left(p\right)/C_{G}$ is dominating and large charge carrier densities even in the order of 10${}^{14}$ cm${}^{-2}$ can be accumulated. Such a high density of itinerant charge carriers has a significant impact on the single-particle level as well as on exciton species, making an unambiguous prediction of the gate-dependent evolution of the electronic and optical properties difficult. Emission spectra in dependence of a wide range of gate voltages swept from 13 V $\leq V_{g}\leq$ −13 V are plotted in Figure 1c. The emission intensity is represented in a false-color representation with high intensities displayed in yellow and low intensities in light violet, respectively. At $V_{g}=$ 0 V a bright emission line PL1 is centered around $E_{PL}^{1}\left(0V\right)\approx$ 2 eV with a pronounced red tail featuring a substructure as will be discussed in more detail below. We note that the PL signature is corrected by a weak fluorescent background from the PE structure as displayed for the individual spectra in Appendix A. PL1 remains bright and slightly redshifts with increasing gate voltage between 0 V $\leq V_{g}\leq$ 7.5 V. The whole PL band significantly redshifts for $V_{g}<$ 0 V and $V_{G}>$ 7.5 V by transitioning from regimes II to regimes III for electron and hole doping, respectively. The redshift is particularly strong for electron doping ($V_{g}<0$) and weaker for hole doping ($V_{g}>0$). Moreover, the intensities reduce rapidly and vanish almost for high negative or high positive gate voltages [regime III($p^{+}/n^{+}$)]. The integrated intensities are plotted as a function of the effective gate voltage as inset in Figure 1c for the down-sweep from 13 V to −13 V and up-sweep from −5 V to 13 V and for comparison for two other samples in Appendix A. The observed and expected hysteretic behavior due to trapped charges and slow ions in the PE is corrected [40]. More information on the gate hysteresis and the corresponding spectra are provided in Appendix A. We assign the gate voltage dependency of the PL spectra and in particular the local maxima in intensity around 0 V and 7.5 V to Fermi-edge singularities, when the chemical potential μ is aligned with the CB edge for $V_{g}\approx$ 0 V and with the VB edge for $V_{G}\approx$ 7.5 V. The intensity increase is interpreted as a result of an increased density of states close to the band edges. The absolute voltage required to shift the chemical potential μ from the VB to the CB is within the uncertainty nearly the same for the up- and down-sweep [inset of Figure 1c].

Individual PL spectra are plotted in a waterfall representation in Figure 2a from the large *n*-doped regime III($n^{+}$) to an intermediate *p*-doped regime II($p^{+}$) from another gate sweep (different doping level at $V_{g}=$ 0 V compared to spectra in Figure 1 due to gate hysteresis). This additional measurement series confirms the overall trend that the main emission band is continuously redshifted in regime I and the overall intensities drop drastically for large electron and hole doping. A rich substructure is evident by displaying the individual spectra. A careful lineshape analysis reveals that the spectra can be well reproduced by a sum of four Gaussians (PL1-PL4) in the doped and highly doped regimes II and III, respectively, and by a sum of three Gaussians in the intrinsic (undoped) regime I. Exemplary spectra together with the individual Gaussians are shown for 12.6 V [regime II($p^{+}$)], 7.2 V [regime I] and 1.4 V [regime II($n^{+})$] in Figure 2b–d.

Additional vdW structures consisting of WS${}_{2}$ MLs encapsulated either on both sides with hBN or by hBN on one side and a few-layer graphene on the other side are prepared and studied in order to disentangle the individual contributions to the PL spectra and to assign them to charged excitons or interfacial disorder. The hBN encapsulation is known to suppress lineshape broadening due to interfacial inhomogeneities [45]. The conducting graphene layer is reported to suppress trion emission by draining free charge carriers [46]. By comparing the deconvoluted emission spectra from the PE-gated WS${}_{2}$ with hBN and graphene interfaced WS${}_{2}$ layers it is evident that emission line PL3 is absent for the graphene supported layer [Figure 2e] and also absent for the PE gated sample for a gate voltage of 7.2 V [regime I] with the $E_{F}$ deep inside the band gap of WS${}_{2}$ [Figure 2c]. This leads to the conclusion that the emission line PL3 is due to charged excitons while emission lines PL1 and PL2 originate from charge neutral excitons. The fact that PL4 is absent in the hBN and graphene interfaced WS${}_{2}$ ML, but observable in the PE gated WS${}_{2}$ in the whole gate voltage range and that PL4 is nearly independent from the gate-voltage suggests that the emission line PL4 is caused by interfacial inhomogeneities and excitons localized to trap states [45].

### 3.3. Doping-Dependent Dielectric Properties and Rydberg States

In order to develop an understanding of the origin of the multiplet emission signatures found for gated WS${}_{2}$ MLs and to determine values for $E_{bind}$ and $E_{gap}$ by measuring the exciton Rydberg series [8], SIE measurements on the WS${}_{2}$-FET structures with a thinner PE electrode are carried out. In Figure 3a, spectra of the measured ellipsometric angles Ψ and Δ (dots) are plotted for the unbiased case. The solid lines are the corresponding fits to the measured data from regression analysis using a multilayer model [41] and modeling the layer structure in a consecutive manner (Appendix A). Clear signatures for the *A* and *B* excitonic ground states $A_{1s}$ and $B_{1s}$, respectively, and weaker but still-observable signatures for the first excited $A_{2s}$ excitonic Rydberg state and the higher lying *C*-excitonic band caused by band nesting [8,41] are apparent in the bare spectra, even though the spectral evolution of the ellipsometric angles is a convolution of all contributing layers. Their spectral positions are indicated by the grey shaded regions in Figure 3a. In our experience, all as prepared (unbiased) WS${}_{2}$-PE structures are moderately *n*-type doped [corresponding to regime II(n)] as also present in the data shown in Figure 1 and Figure 2.

The approach and the multilayer model used for fitting the ellipsometric data are identical for the unbiased and biased measurements. The multilayer model consists of glass substrate, interlayer, WS${}_{2}$ ML, thin film PE electrode and air. As expected, only the dielectric function of WS${}_{2}$ is strongly dependent on the gate voltage [47]. The dielectric functions and equivalently refractive index n(E) and extinction coefficient κ(E)-with the latter being directly proportional to the absorption α(E)-are extracted from the multilayer model for all investigated gate voltages. The extinction coefficient κ(E) for increasing *n*-type doping is plotted in Figure 3b for unbiased, $V_{g}=$ −1 V and $V_{g}=$ −4 V corresponding to slightly *n*-doped [II(*n*)], moderately *n*-doped [III($n^{+}$)] and highly *n*-doped [III($n^{++}$)] regimes, respectively. In regimes II and III, we assume the change in the charge carrier density to be in the order of Δn(1V$)\approx 1.25\times 10^{13}$ cm${}^{-2}$ per applied volt between gate-electrode and WS${}_{2}$ as defined in the method section. We find that with increasing electron density *n*, the $A_{1s}$ exciton resonance broadens and completely vanishes for $V_{g}=-4\;$V$\left(n^{++}\right)$ similar to the excited $A_{2s}$ state that is already unresolvable at $V_{g}=-1\;$V$\left(n^{+}\right)$ indicating the dissociation of the *A* exciton. The width of the $B_{1s}$ exciton resonance is less affected for a moderate increase of the electron density [47], but vanishes as well for high electron doping $n^{++}$. Only the *C*-exciton band remains nearly unaffected from doping, which is not surprising since it originates from higher-lying interband excitations with contributions from states between Γ and *M* point that are far away from the occupied CB states [41]. The observed evolution of the absorption spectra with electron doping can be well understood in terms of the interaction of the bound exciton states with the increasing density of mobile electrons that screens the attractive Coulomb interaction [48]. For moderate doping, the formation of Mahan excitons, “dressed” excitons (exciton in a Fermi see) and polarons is discussed in literature [49,50,51,52]. For high and highest doping [III($n^{++}/n^{+},p^{++}$)], a Mott-transition to a weakly interacting electron and hole plasma is expected [48]. The occurrence of the Mott-transition in the highly doped regime is debated in literature [49]. Here, we assume the absence of *A* and *B* excitons resonances in the highly n-doped regime [III($n^{++}$)] as clear indication for transition into the Mott-regime [21,33].

For positive gate-voltages 0 V $\leq V_{g}\leq$ 12 V, signatures for the exciton ground states $A_{1s}$ and $B_{1s}$ as well as for the first excited exciton state $A_{2s}$ are clearly visible in the ellipsometry spectra and well quantified in the extracted dispersion function from regression analysis. The extracted energies for the three lowest exciton signatures as a function of gate-voltage (mainly regime I and partially covering regime II) are summarized in Figure 3c. The separation between $A_{1s}$ and $B_{1s}$ excitons is independent of the applied gate voltage and constitutes about 370 ± 20 meV. It originates from the spin-orbit split valence bands at the *K*, $K^{\prime }$ points and agrees well with [48]. There is a slight redshift of the *A* and *B* exciton energies of about 40 meV with increasing gate voltages from 0 V to 12 V that covers regimes I and II(n/p). A similar redshift has been observed for the main emission line [see Figure 1c]. From the energy difference $\Delta E_{12}$ between the exciton ground state $A_{1s}$ and the first excited state $A_{2s}$, the exciton binding energy can be deduced from the hydrogenic Rydberg series considering a finite degree of nonlocal screening of the attractive Coulomb interaction between electron and hole because of inhomogeneous dielectric environment normal to the 2D layer. Assuming fully non-hydrogenic scaling, the binding energy constitutes $E_{bind}=2\cdot \Delta E_{12}$ and for hydrogenic scaling $E_{bind}=9/8\cdot \Delta E_{12}$ providing an upper and lower bound for $E_{bind}$. [19,48]. The actual value for $E_{bind}$ is expected within these boundaries and depends on the dielectric screening of substrate and capping PE layer as well as on the doping density in the 2D semiconductor [19]. In the present case, the maximum in the binding energy is found for $V_{g}=$ 4 V to be in the range between $E_{bin}$ = 0.5 V and 0.25 eV for scaling factors 2 and 9/8, respectively [c.f. Figure 3d]. The values for the single-particle band gap $E_{gap}$ can then be deduced from the relation $E_{gap}=E_{A_{1s}}+E_{bind}$. We find that the binding energy first increases from 0 V to ≈4 V and decreases again for larger gate voltages as depicted in Figure 3d. These findings allow for the conclusion that for the investigated structure the chemical potential and, hence, $E_{F}$ is centered in the middle of the band gap of WS${}_{2}$ for $V_{g}\approx$ 4 V in agreement with emission spectroscopy. $E_{F}$ approaches the CB edge for lower gate voltages and the VB edge for larger voltage such that screening is already slightly increased by thermally excited charge carriers as well as by initially injected carriers. The extracted gap energies are 2.27 ± 0.1 eV and 2.47 ± 0.1 eV, respectively. The binding energy is reduced by about 15% for slight n-doping and reduced up to about 35% for moderate p-doping in the experimentally accessible range. For higher doping, the $A_{2s}$ exciton is no longer well-resolved in SIE measurements.

## 4. Discussion

We start the discussion with a classification of the different PL emission lines. The gate voltage-dependent integrated PL intensities, energies of the emission lines PL1, PL2, PL3 from the deconvolution with Gaussian profiles [c.f. Figure 2] as well as the relative peak intensities are summarized in Figure 4a–c. The measurements are done by sweeping the gate voltage from −6 V $\leq V_{g}\leq$ +13 V [c.f. upsweep in inset of Figure 1c]. The energies of the absorption peak $A_{1s}$ from SIE investigation are included in Figure 4b even though the voltage-induced charge carrier densities are expected to slightly differ, since the measurements are done on different samples. The sweep direction of the gate voltage was identical across the measurements to reduce the impact of the hysteretic effect of the PE gate electrode. We would like to note that the position of $E_{F}$ for the unbiased case changes moderately from sample to sample, allowing only a qualitative comparison. Nevertheless, the assignment to different regimes is reliable and is nicely confirmed by the gate dependence of the dielectric properties. While showing nearly the same gate voltage dependence, the energies of $A_{1s}$ and PL1 differs independent from the applied gate voltage $V_{g}$ by about 30 meV. This energy difference is robust and has been confirmed by PL and SIE measurements on several samples with and without PE gate electrode. By cooling the WS${}_{2}$ ML to 77 K, also a weak blue-shifted emission line appears about 30 meV from the main emission line PL1 (see Appendix A). The peak energies are compared in Table 1 for specific gate voltage values corresponding to $E_{F}$ centered in the middle of the band gap [I], $E_{F}$ aligned with CB and VB edges [II(n/p)] and for high electron and hole doping $n^{+}/p^{+}\approx$$5\times 10^{13}$ cm${}^{-2}$ [III($n^{+}/p^{+}$)]. There are no values for $A_{1s}$ in the high $n^{+}$- and $p^{+}$-doped regions because of the screening induced weak absorption intensities. The nearly gate voltage-independent energy differences between $A_{1s}$ and PL1 constitute 30 to 40 meV and between PL1 and PL2 about 40 meV. Emission line PL3 exhibits a significantly different behavior and vanishes if $E_{F}$ is centered in the band gap. The energy difference between PL2 and PL3 increases with increasing *p* and *n* doping up to 30 meV for doping densities of about ($n^{+}/p^{+}\approx$$5\times 10^{13}$ cm${}^{-2}$).

The fact that the observed emission lines PL1-PL3 do not appear in the absorption spectra determined from SIE and are redshifted in energy with respect to the $A_{1s}$ peak is a strong evidence for their spin or momentum indirect nature. This points to the involvement of phonons in the recombination process of the associated excitons. Moreover, PL3 is absent in the emission spectra for the charge-neutral situation either realized by gate-tuning $E_{F}$ inside the band-gap [regime I] (Figure 4) or by using supporting graphene as a trion-filtering layer [46] (c.f. Figure 2). The interpretation of PL3 as trion is supported by the fact that its relative intensity is significantly enhanced in regimes II(*n*) and II(*p*) and becomes the most intense emission line for increasing doping densities in regimes III($n^{+}$) and III($p^{+}$) [Figure 4c].

We now turn to the interpretation of the two remaining charge-neutral phonon-assisted emission lines, PL1 and PL2. At the lowest energy direct interband transition at the *K*, $K^{\prime }$ points, the lowest CB and VB states have opposite spins (see sketch in Figure 5). This optical interband transition is therefore spin-forbidden and, hence, dark excitons are generated by optical excitation that can only recombine by emitting or absorbing a suitable phonon connecting *K* and $K^{\prime }$ valleys in order to mediate momentum conservation. The energy difference between the spin-forbidden “dark” state and spin-allowed “bright” transition from the highest VB state to the spin-split CB state at the *K*, $K^{\prime }$ valleys is about 30 meV [48]. This nicely coincides with the energy separation between $A_{1s}$ and PL1. Following this argumentation, PL1 is a phonon-assisted emission from an exciton formed between the spin-split CB and the VB at the $K^{\prime }/K$ points. The remaining peak PL2 is interpreted as phonon-activated transition between the Σ-valley in the CB and the *K*-valley in the VB. The interpretation that all emission bands observed in room temperature PL experiments are phonon-activated spin/momentum indirect transitions are corroborated by the spectrum of the extinction coefficient κ(E) (Figure 3c) and low-temperature PL measurements (Appendix A) and in agreement with some experimental and theoretical reports in the literature [17,26,36,53].

By tuning $E_{F}$ from the CB edge through the band-gap to the VB edge, the peak energies of the prominent charge-neutral excitonic features $B_{1s}$, $A_{1s}$, PL1 and PL2 are monotonously redshift by about 35 ± 5 meV between $E_{CB}$ and $E_{VB}$ (see Figure 1c, Figure 2a, Figure 3b and Figure 4b). There is either no or only weak charge doping in the relevant regime I, so that we do not expect doping-induced band renormalization phenomena. The exciton binding energies of the $A_{1s}$ state in this regime are first slightly increased until $E_{F}$ is centered in the band-gap and is then again reduced (Figure 3d). We assume that the monotonous reduction of the transition energies with gate voltage is caused by the different action of the electric field on the orbital contribution for the relevant CB and VB states [53].

The reduction of the peak energies of PL1, PL2 and PL3 emission lines transitioning from the weakly doped II(*n*) regime into the highly electron-doped III($n^{+}$) regime can be explained by a combination of doping induced band renormalization and screening effects [53]. Band renormalization effects are highly sensitive to the orbital contributions of the Bloch states at the different high-symmetry points in the 1st Brillouin Zone (BZ) [53]. Assuming *k*-dependent renormalization, we can explain that the PL2 emission energies redshift already at a gate voltage of $V_{g}\approx$ 1.4 V, while for PL1 the redshift is less pronounced and starts at $V_{g}$ = 1 V in the data set analyzed in Figure 4 because for PL2 the the electrons are hosted in the Σ-valley, while for PL1 in the *K*–valley. Since those emission lines are phonon activated, doping induced phonon renormalization and altered electron–phonon interaction might serve as an additional factor for the gate-dependent evolution of the emission spectra [38,54].

The evolution of the emission energies for larger electron doping with increasing negative gate voltages can be understood in terms of compensation of doping-induced band renormalization and Coulomb screening lowering $E_{gap}$ and $E_{bind}$ such that the optical transition energies $E_{opt}=E_{gap}-E_{bind}$ remains nearly constant within the experimental resolution [24]. For larger doping densities [III($n^{++}$)], the doping-induced screening results in a Mott transition to an electron and hole plasma. This is in agreement with the experimental observation that the PL emission intensity is drastically reduced.

## 5. Conclusions

In conclusion, we successfully demonstrate optically detected capacitance spectroscopy of CB and VB edges of WS${}_{2}$ MLs realized by a solid-electrolyte gate field-effect structure. The exciton binding $E_{bind}$ and the single-particle gap $E_{gap}$ energies determined by analyzing the excitonic Rydberg series from SIE investigations are maximized for $E_{F}$ roughly entered inside the band gap (undoped situation) and constitute $E_{bind}\approx$ 0.5 eV and $E_{gap}\approx$ 2.5 eV, respectively. The method is applicable to other optically active 2D materials. Moreover, we demonstrated ambipolar tunability of the charge carrier density in WS${}_{2}$ ML from high *n*-type to a high *p*-type doping, exceeding the doping level required to induce exciton Mott transition. The gate voltage-dependent study of both PL and SIE measurements allows us to assign the dominant emission features to spin- and momentum-forbidden transitions and provide further evidence for this interpretation in comparison with PL spectra from graphene/WS${}_{2}$/hBN and hBN/WS${}_{2}$/hBN structures. At low charge carrier densities and for the undoped situation, these spin and momentum indirect transitions dominate the optical response. We find that for $E_{F}$ close to the CB and VB edges [regime II(n/p)], charged excitons also contribute to the emission spectrum, while in highly doped regimes the emission signatures are interpreted to originate from an unbound electron hole plasma. In this context, SIE measurements have shown that strong exciton resonances occur for regimes [I] and [II(n/p)], while large charge carrier densities result in screening induced reduction and disappearance of exciton resonances in the absorption expressed by the extinction coefficient κ(E). Our work thus highlights the rich physics and tunability of the optical properties of WS${}_{2}$ ML embedded in electrolyte-gated field-effect structures. This platform is of high relevance for both fundamental understanding and its potential for implementation in quantum devices.

## Figures and Tables

**Figure 1 nanomaterials-12-03255-f001:**
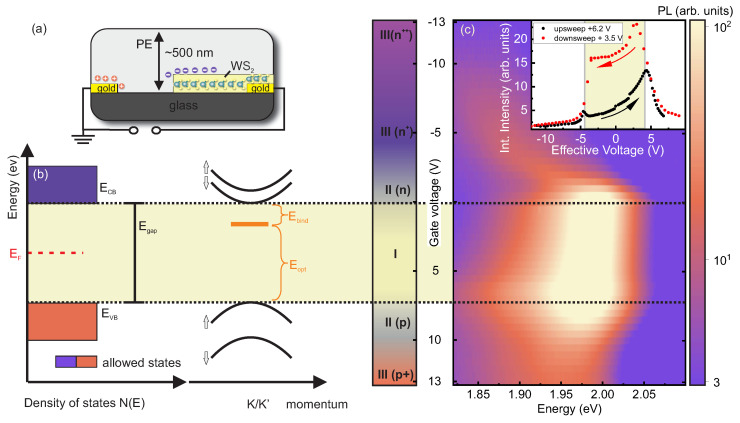
(**a**) Sketch of the 1 L WS${}_{2}$ field-effect structure using a solid electrolyte gate electrode. (**b**) Schematic density of states N(E) and the single-particle band structure close to the fundamental band-gap for a 2D semiconductor. Since the electrochemical potential μ of the gate electrode is aligned with the Fermi level $E_{F}$ of the WS${}_{2}$ semiconductor, $E_{F}$ can be tuned by an applied gate voltage. (**c**) PL spectra in a false-color representation for varying gate voltages $V_{g}$ 13 V to −13 V. The emission intensity is color-coded. When the chemical potential and, hence, $E_{F}$ is aligned with the CB and VB edges, an enhanced emission intensity occurs. Intensity and lineshape is interpreted as interplay of screening and doping dependent behavior of the rich exciton manifolds in WS${}_{2}$ MLs. Inset: Integrated intensities for gate up and down sweeps (hysteresis-corrected). [*T* = 300 K, $\lambda_{Laser}$ = 532 nm; *P* = 50 μW].

**Figure 2 nanomaterials-12-03255-f002:**
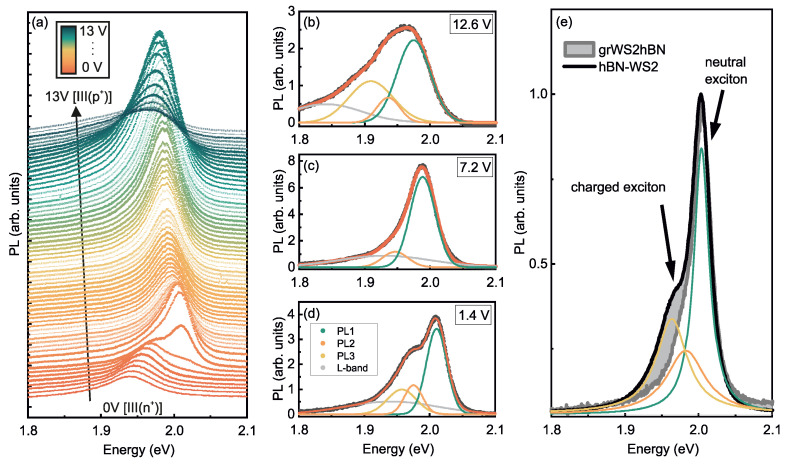
(**a**) Water Fall representation of individual spectra for a gate voltage sweep from 0 V to 13 V clearly displaying changes in intensity and lineshape. (**b**–**d**) Line-shape analysis using a sum of 3 or 4 Gaussian reproducing the spectra for gate voltages of 12.6 V [II(*p*)] (**b**), for 7.2 V [I] (**c**) and of 1.4 V. [III($n^{+}$)] (**d**). Emission line PL3 is absent in the intrinsic regime. PL4 is assigned to defect related emission. (**e**) PL spectra for a WS${}_{2}$ ML encapsulated in hBN (green trace) and in hBN and few-layer graphene (red trace). The graphene layer drains free charge carriers such that only charge-neutral excitons contribute to the emission. The spectra reproduced by 2 or 3 Gaussians such that PL3 is assigned to trion and PL4 to defect emission (absent in encapsulated WS${}_{2}$). [*T* = 300 K, $\lambda_{Laser}$ = 532 nm; *P* = 50 μW].

**Figure 3 nanomaterials-12-03255-f003:**
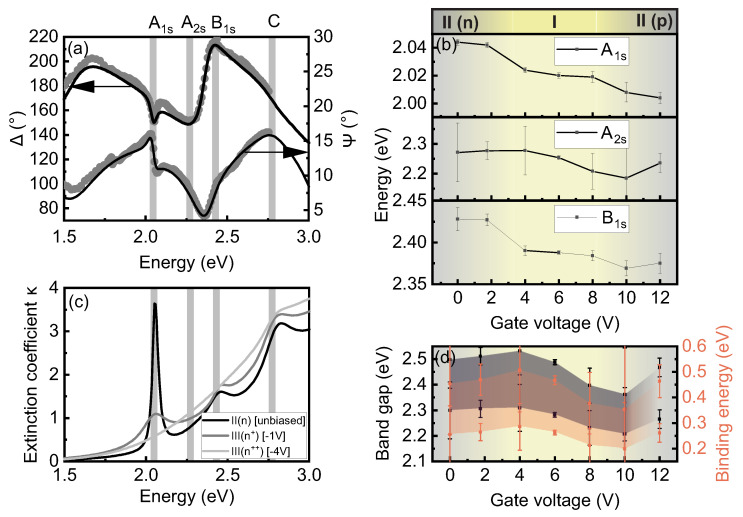
(**a**) Spectra of the ellipsometric angles Δ and Ψ for WS${}_{2}$–PE gate field-effect structure for the unbiased case (symbols). Solid line represents the fit to the model from regression analysis. The excitonic signatures are indicated. (**b**) Extracted $A_{1s}$, $A_{2s}$ and *B* exciton energies in dependence of positive gate voltages 0 V $\leq V_{g}\leq$ 12 V moving $E_{F}$ from the CB into the band gap to the VB edge. [*T* = 300 K]. (**c**) Extinction coefficient κ(E) extracted from SIE for unbiased, $V_{g}=$ 1 V and $V_{g}=$ −4 V corresponding to weak [II(*n*)], moderate [III($n^{+}$)] and high electron doping [III($n^{++}$)], respectively. $A_{1s}$, $A_{2s}$, *B* and *C* exciton signatures are indicated. (**d**) $V_{g}$-dependent evolution of exciton binding energies determined from the energy difference between $A_{1s}$ and $A_{2s}$ (red) and single-particle band gap (black) considering nonlocal screening [upper and lower limit as explained in the text].

**Figure 4 nanomaterials-12-03255-f004:**
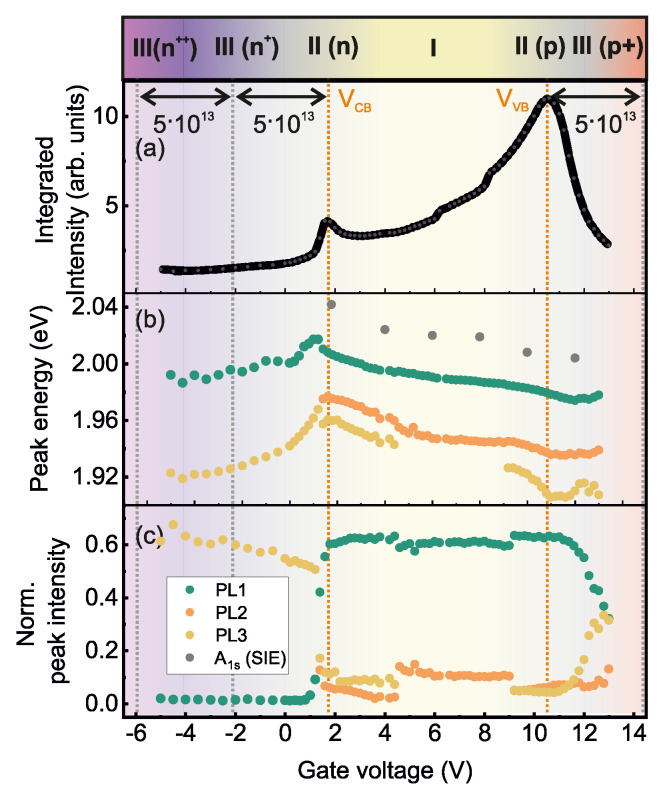
(**a**) Integrated emission intensities as a function of gate voltages from −5 V $\leq V_{g}\leq$ 13 V from the spectra displayed in Figure 2a. The maxima indicate voltage values for which $E_{F}$ is aligned to CB and VB edges. The different doping regimes are highlighted. (**b**) Peak energies of the line-shape analysis using Gaussian profiles as a function of gate voltages for emission lines PL1, PL2 and PL3. The grey circles are the $A_{1s}$ energies from SIE measurements. Since the measurements were done on two different samples, the effective electric field transferred by the gate voltage is expected to slightly differ. (**c**) Intensities of the emission lines shown in (**b**) are normalized to the integrated emission energies shown in (**a**).

**Figure 5 nanomaterials-12-03255-f005:**
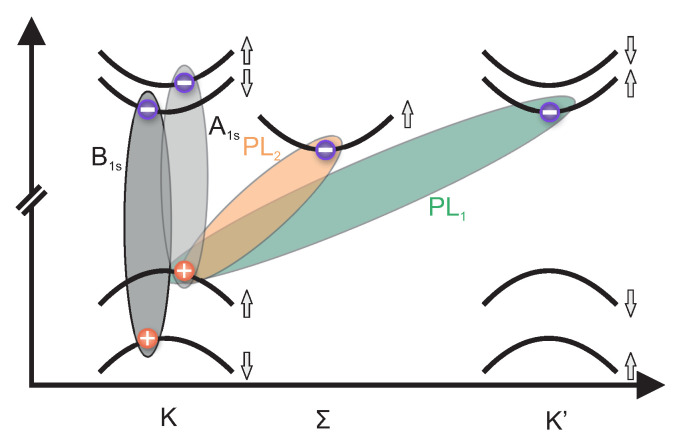
Schematic representation of the exciton formed by holes (red spheres) residing in the *K* valley and electrons (blue spheres) residing in *K*, K′ and Σ valleys that are interpreted to originate $B_{1s}$, $A_{1s}$, PL1 and PL2, respectively.

**Table 1 nanomaterials-12-03255-t001:** Energies of exciton species found in PL and SIE measurements for selected doping concentrations and positions of the Fermi-energy $E_{F}$. The uncertainty in the determination of the energies is in the order of 10 meV.

$E_{F}$/Doping	$A_{1s}$ (eV)	PL1 (eV)	PL2 (eV)	PL3 (eV)
$n^{+}(\approx$ 5$\times 10^{13}$ cm${}^{-2}$)	-	1.99	1.95	1.92
$E_{F}\approx E_{CB}$	2.04	2.01	1.975	1.96
$E_{F}\approx 0$	2.03	1.99	1.95	-
$E_{F}\approx E_{VB}$	2.01	1.98	1.94	1.92
$p^{+}(\approx$ 5$\times 10^{13}$ cm${}^{-2}$)	-	1.98	1.94	1.91

## Data Availability

The data presented in this study are available on request from the corresponding author.

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
