# Peer review of "Exciton Manifolds in Highly Ambipolar Doped WS2"

_nanomaterials, 2022, doi:10.3390/nano12183255_

Round 1
Reviewer 1 Report
Authors study the optical properties of WS2 flakes doped by ion-gating, an interesting topic for the development of optoelectronic devices. Here some key points missing in the manuscript that should be addressed:
-a photograph of the device is not shown, and it will help to have an idea of the devices’ dimensions and geometry, both relevant for the device performance.
-authors should indicate “power density values” in the experimental section.
-authors should explain the way they determine the charge carrier densities indicated in the manuscript, and therefore how they established the regimes. From the text, it seems that the values are taken from literature.
-PL4 fitting is not indicated in Figure 2, Reviewer supposes that it should be the grey line.
-authors should include a statement regarding reproducibility, since they just comment “minor variations are expected from sample to sample.” They can use the SI for including experimental data from other devices.
-check the spelling along the text, there are several typos such as “theses”
Reviewer 2 Report
This paper discusses a establish an electrolyte gated WS2 monolayer field-effect structure, which can shift the Fermi level from the valence into the conduction band. They found that this field-effect platform is suitable for studying exciton manifold and combined optical and transport measurements on thin quantum materials at cryogenic temperatures. The characterization and the discussion are sufficient. However I question the novelty and fundamental difference from more literature.
1. In the experiment part, why 5nm chromium is selected as adhesion layer? Is this layer result in energy mismatch between Cr and the WS2?
2. Line 140~143, the author declares the hysteresis during measurement. How to reduce this hysteresis?
Reviewer 3 Report
The comments for the manuscript are shown in a separate file.

Round 2
Reviewer 1 Report
Authors have addressed the comments from the Reviewers.